

# Lung–brain 'cross-talk': systemic propagation of cytokines in the ARDS *via* the bloodstream using a blood transfusion model does not influence cerebral inflammatory response in pigs

René Rissel, Moritz Schaefer, Jens Kamuf, Robert Ruemmler, Julian Riedel, Katja Mohnke, Miriam Renz, Erik K. Hartmann and Alexander Ziebart

Department of Anaesthesiology, Medical Centre of the Johannes Gutenberg-University, Mainz, Germany

Corresponding author
René Rissel,
Rene.Rissel@unimedizin-mainz.de

## ABSTRACT

**Background:** Interorgan cross-talk describes the phenomenon in which a primarily injured organ causes secondary damage to a distant organ. This cross-talk is well known between the lung and brain. One theory suggests that the release and systemic distribution of cytokines *via* the bloodstream from the primarily affected organ sets in motion proinflammatory cascades in distant organs. In this study, we analysed the role of the systemic distribution of cytokines *via* the bloodstream in a porcine ARDS model for organ cross-talk and possible inflammatory changes in the brain.

**Methods:** After approval of the State and Institutional Animal Care Committee, acute respiratory distress syndrome (ARDS) induction with oleic acid injection was performed in seven animals. Eight hours after ARDS induction, blood ($35–40 \text{ ml kg}^{-1}$) was taken from these seven 'ARDS donor' pigs. The collected 'ARDS donor' blood was transfused into seven healthy 'ARDS-recipient' pigs. Three animals served as a control group, and blood from these animals was transfused into three healthy pigs after an appropriate ventilation period. All animals were monitored for 8 h using advanced cardiorespiratory monitoring. Postmortem assessment included cerebral (hippocampal and cortex) mediators of early inflammatory response (IL-6, TNF-alpha, iNOS, sLCN-2), wet-to-dry ratio and lung histology. TNF-alpha serum concentration was measured in all groups.

**Results:** ARDS was successfully induced in the 'ARDS donor' group, and serum TNF-alpha levels were elevated compared with the 'ARDS-recipient' group. In the 'ARDS-recipient' group, neither significant ARDS alterations nor upregulation of inflammatory mediators in the brain tissue were detected after high-volume random allogenic 'ARDS-blood' transfusion. The role of the systemic distribution of inflammatory cytokines from one affected organ to another could not be confirmed in this study.

## INTRODUCTION

Brain and lung injuries are some of the most common causes of patient admission to the intensive care unit (*Pelosi & Rocco, 2011*; *Brower et al., 2000*). The management of these patients is complex, and high rates of morbidity and mortality are associated with these illnesses. Injuries to the brain or lung result in inflammatory responses, which frequently lead to secondary injuries in remote organs. This phenomenon is referred to as organ cross-talk, and it describes the interactions between a primary affected organ and a secondarily injured remote organ. Interorgan signalling networks are not only described for brain–lung cross-talk, but for other organ pairs as well. The dysregulations of these interorgan communication networks are also known in renal and metabolic diseases (*Ologunde et al., 2014*; *Priest & Tontonoz, 2019*). The remote organ is primed and more sensitive to other stimuli that can cause organ damage (*e.g.*, infections). The progression of brain injury to acute lung injury because of the excessive release of cytokines and other neurohumoral factors is often explained as a 'double hit' (*Mrozek, Constantin & Geeraerts, 2015*). Patients who have survived ARDS often develop some degree of cognitive deterioration, which has long-term effects on the brain because of the intensive therapy required (*Hopkins & Jackson, 2006*). A common theory is that there is a systemic distribution of inflammatory mediators released from the affected organ and transmitted *via* the bloodstream to other organs, such as the brain (*Mrozek, Constantin & Geeraerts, 2015*; *Ziebart et al., 2019*; *Klein et al., 2016*; *Hegeman et al., 2009*). Three transmission pathways of lung–brain cross-talk have been described: a humoural, a cellular and a neuronal pathway (*López-Aguilar et al., 2013*). Furthermore, dysregulations of the hypothalamo-pituitary-adrenal axis contribute to changes in the stress and inflammatory response after lung injury (*Munford & Pugin, 2001*). At the very least, alterations in blood–brain barrier's (BBB) function contribute to a local inflammatory response in the central nervous system (CNS) (*Varatharaj & Galea, 2017*). In the past, little attention has been paid to brain–lung cross-talk (*Pelosi & Rocco, 2011*). Hence, the present study investigates the role of the systemic distribution of cytokines *via* the bloodstream in a porcine ARDS model for organ cross-talk and possible inflammatory changes in the brain.

## MATERIALS AND METHODS

### Anaesthesia and instrumentation

This prospective randomised animal study was conducted after approval by the State and Institutional Animal Care Committee (Landesuntersuchungsamt Rheinland-Pfalz, Mainzer Straße 112, 56068 Koblenz, Germany; reference number: 23 177-07/G 14-1-084), here in accordance with the ARRIVE guidelines (*Kilkenny et al., 2010*).

The anaesthesia and instrumentation are highly standardized in our group and were previously reported (*Ziebart et al., 2019*; *Kamuf et al., 2018*). To minimise stress, the animals (*Sus scrofa* domestica) stayed in their familiar environment for as long as possible. A local breeder took care of their general condition. After intramuscular injection of ketamine (8 mg kg$^{-1}$) and midazolam (0.2 mg kg$^{-1}$) the sedated animals were delivered to the laboratory. After supine positioning, fentanyl (4 μg kg$^{-1}$) and propofol (4 mg kg$^{-1}$)

were injected intravenously to induce general anaesthesia. Throughout the entire experiment, anaesthesia was maintained through the continuous infusion of fentanyl ($0.1–0.2\ \mu g\ kg^{-1}\ h^{-1}$) and propofol ($8–12\ mg\ kg^{-1}\ h^{-1}$). A single dose of atracurium ($0.5\ mg\ kg^{-1}$) was administered to facilitate endotracheal intubation. The animals were ventilated in volume-controlled mode (AVEA; CareFusion, San Diego, CA, USA): tidal volume $8\ ml\ kg^{-1}$; positive end-expiratory pressure (PEEP) 5 cm $H_2O$, fraction of inspired oxygen ($FiO_2$) 0.6, inspiration to expiration ratio 1:2 and variable respiration rate to achieve an end-tidal $P_{CO_2}$ < 6 kPa. Under ultrasound guidance, four femoral vascular catheters were placed as follows: central venous line (drug administration); pulse contour cardiac output system (PiCCO; Pulsion Medical Systems, Munich, Germany); arterial line (blood withdrawal); and large-bore venous introducer (fluid resuscitation). The haemodynamic and spirometric parameters were permanently measured and recorded (S5; GE Healthcare, Chicago, IL, USA). The transpulmonary thermodilution technique with single indicators was used to measure cardiac output, global end-diastolic volume and extravascular lung water. To analyse the regional ventilation distribution, we used an electrical impedance tomography device (EIT; Goe-MF II, CareFusion, San Diego, CA, USA), which records thoracic bioimpedance variations associated with tidal ventilation. Electrodes were placed on a transverse lung section just below the axilla. Regional ventilation distribution was examined for the nondependent, central and dependent lung areas (Levels L1–L3) as a percentage of the global tidal amplitude.

## Experimental design

The experimental protocol is displayed in Fig. 1. After instrumentation and a 30 min consolidation, baseline measurements were documented, and the ARDS was induced with oleic acid injection ($0.1\ ml\ kg^{-1}$, Applichem GmbH, Darmstadt, Germany) until the partial pressure of oxygen/fraction of inspired oxygen ratio ($PaO_2/FiO_2$) was below 200. This is a common ARDS model in pigs, as reported by *Kamuf et al. (2018)* Blood samples, EIT measurements and spirometric parameters were taken at baseline and after ARDS induction at 4 and 8 h. After 8 h, blood sampling was performed ($35–40\ ml\ kg^{-1}$) in seven animals (Group 1, 'ARDS donor', $n = 7$). Blood was collected from the donor animals *via* the arterial catheter for allogeneic blood transfusion. Whole blood was collected in a specific collector bag system (Composelect; Fresenius-Kabi AG, Homburg, Germany). These bags contained 63 ml CPD 100 $ml^{-1}$ SAG-M-RCC (Citrate phosphate dextrose, erythrocyte storage in hypertonic conservation medium). After collection in a primary bag, the blood was drained through a leucocyte-depleting filter into a second bag containing the anticoagulants. Finally, the whole blood was collected into a bag with a port for a transfusion system. The collected blood was stored cool and protected from light at 4 °C for 12 h. At the end of observation, the animals were euthanised during deep general anaesthesia by central venous administration of propofol and potassium chloride. The organs were removed from the dead animal for further examination. The following day, the recipient animal was prepared as described above, and the collected stored blood was transfused after 30 min of baseline time at 0 h. Blood samples, EIT measurements and spirometric parameters were also taken at baseline, after blood transfusion, after 4 h
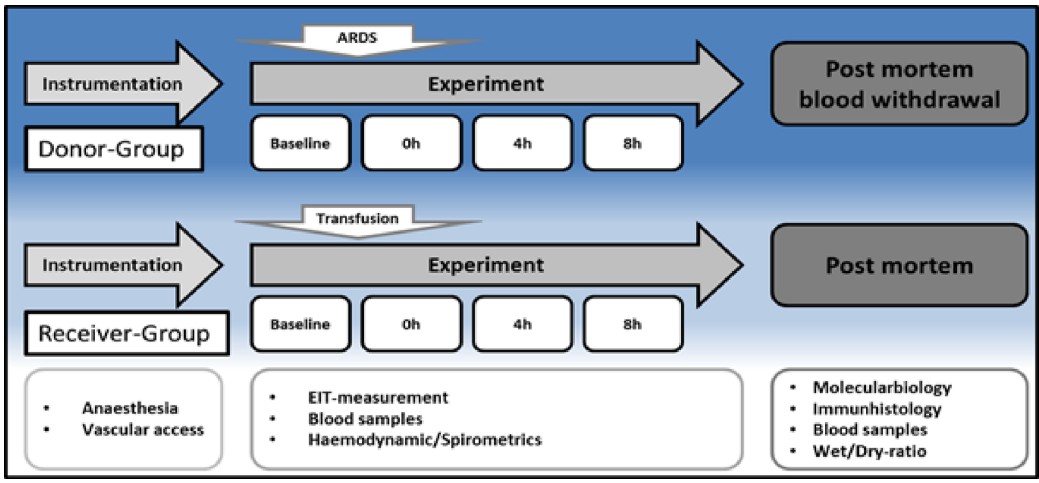

**Figure 1 Timeline.** H, hours; EIT, electrical impedance tomography; ARDS, acute respiratory distress syndrome.

and after 8 h. These animals were also euthanised during deep general anaesthesia by central venous administration of propofol and potassium chloride. The organs were removed from the dead animal for further examination. The animals in Group 2 (donor (s) = donor-sham, common blood transfusion, $n = 3$) were prepared as the animals in the group without the induction of an ARDS with oleic acid. After 8 h of the experiment, blood withdrawal was performed, as described above, and transfused on the following day (receiver (s) = receiver-sham, $n = 3$).

## Postmortem analysis

Repeatedly collected serum blood samples were used to determine the serum levels of tumor necrosis factor-alpha (TNF-alpha) enzyme-linked immunosorbent assays (Porcine Quantikine ELISA Kits; R/D Systems, Wiesbaden, Germany). At the end of the protocol, the lung and brain were each removed en-bloc. Cortex and hippocampus samples were cryopreserved for an mRNA expression analysis of inflammatory mediators like interleukin-6 (IL-6), tumor necrosis factor-alpha (TNF-alpha), inducible nitric oxide synthase (iNOS) and soluble lipocalin-2 (sLCN-2) by real-time polymerase chain reaction (rt-PCR; Lightcycler 480 PCR System; Roche Applied Science, Penzberg, Germany), as described in detail by *Ziebart et al. (2019)* mRNA expression was used to better compare the results with previous studies and normalised to peptidylprolyl isomerase A (PPIA) (*Ziebart et al., 2019*). To determine the lung damage, we used a standardised scoring system, as described in detail from our group previously (*Ziebart et al., 2014*). Furthermore, the lung water content was determined through the tissue wet-to-dry ratio. For this procedure, the weight of the removed left lung tissue sample was determined directly and after 2 d of complete drying.

## Statistics

Group size was adjusted comparably with recently published studies (*Ziebart et al., 2019*; *Hartmann et al., 2014*). The values are displayed as the mean and standard deviation (SD)

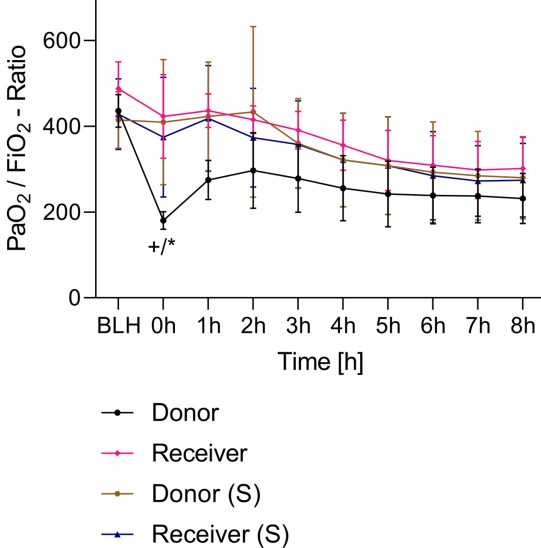

**Figure 2 PaO$_2$/FiO$_2$-ratio.** $+p < 0.001$ donor group 0 h $vs.$ BLH, $^*p < 0.001$ donor $vs.$ recipient/donor (S)/recipient (S) at 0 h. h, hours; BLH, baseline healthy; (S), sham.

or violin plots (median and interquartile range). To analyse the effect between the three groups over time, we performed a two-way analysis of variance (ANOVA) with pairwise multiple comparison correction using the Holm-Sidak method. The Mann–Whitney $U$-test was used to analyse the postmortem parameters. A $p$-value less than 0.05 was accepted as significant. Statistical analysis was performed with the SigmaPlot 12.5 software package (Systat Software, Chicago, IL, USA) and GraphPad Prism 9.3.1 (GraphPad Software, San Diego, CA, USA).

## RESULTS

All animals survived the 8 h experimental period (donor $n = 7$; receiver $n = 7$; donor-sham $n = 3$; receiver-sham $n = 3$). ARDS induction with oleic acid infusion was successfully performed in all animals (donors), with no observed adverse events. The PaO$_2$/FiO$_2$-ratio was significantly reduced after ARDS induction (donor group 0 h $vs.$ baseline healthy (BLH): $436 \pm 35$ $vs.$ $180 \pm 19$; $p < 0.001$). Furthermore, a significant intergroup difference was observed between the PaO$_2$/FiO$_2$ ratio at 0 h (donor $vs.$ donor-sham/receiver/receiver-sham: $180 \pm 19$ $vs.$ $455 \pm 35/423 \pm 90/410 \pm 119$; $p < 0.001$). Figure 2 illustrates the PaO$_2$/FiO$_2$ ratio. In the donor and receiver groups, the mean pulmonary arterial pressure increased after ARDS induction and blood transfusion without being significant. No further significant alterations in the other groups were measured. The key hemodynamic data set is summarized in Table 1 and documents comparable conditions in all groups throughout the trial. The end-diastolic lung water index (ELWI) increased significantly in the donor group after ARDS induction (0 h $vs.$ BLH: $12 \pm 3$ $vs.$ $14 \pm 3$; $p < 0.035$). Furthermore, a demonstrably increased ELWI was measured in the receiver group (8 h $vs.$ BLH: $15 \pm 4$ $vs.$ $11 \pm 2$; $p < 0.001$). As summarized in Table 2, spirometry data showed no intergroup differences. The distribution of regional ventilation showed no

**Table 1 Hemodynamic parameters.**

| Parameter | Group | BLH Mean (SD) | 0 h Mean (SD) | 4 h Mean (SD) | 8 h Mean (SD) |
|---|---|---|---|---|---|
| MAP [mmHg] | Donor | 71 (10) | 73 (7) | 66 (6) | 64 (3) |
| | Donor (S) | 61 (3) | 61 (3) | 66 (2) | 65 (5) |
| | Receiver | 64 (5) | 97 (13) | 80 (12) | 74 (14) |
| | Receiver (S) | 64 (5) | 97 (13) | 80 (6) | 74 (6) |
| HR [min$^{-1}$] | Donor | 79 (12) | 137 (38) | 169 (43) | 135 (28) |
| | Donor (S) | 82 (18) | 82 (18) | 105 (22) | 98 (15) |
| | Receiver | 76 (10) | 87 (14) | 91 (7) | 87 (12) |
| | Receiver (S) | 80 (16) | 96 (11) | 87 (11) | 88 (15) |
| PAP [mmHg] | Donor | 17 (2) | 32 (7) | 20 (5) | 16 (6) |
| | Donor (S) | 14 (3) | 14 (3) | 17 (2) | 16 (3) |
| | Receiver | 12 (4) | 27 (11) | 16 (4) | 15 (4) |
| | Receiver (S) | 19 (7) | 38 (10) | 26 (8) | 24 (7) |
| CO [l min$^{-1}$] | Donor | 3 (0) | 4 (1) | 4 (1) | 5 (1) |
| | Donor (S) | 3 (0) | 3 (0) | 3 (0) | 4 (1) |
| | Receiver | 3 (1) | 3 (1) | 3 (1) | 3 (1) |
| | Receiver (S) | 3 (0) | 3 (1) | 3 (0) | 2 (0) |
| GEDI [ml kg$^{-1}$] | Donor | 468 (78) | 459 (95) | 435 (84) | 458 (85) |
| | Donor (S) | 396 (85) | 396 (85) | 386 (82) | 378 (66) |
| | Receiver | 466 (72) | 497 (93) | 472 (72) | 464 (84) |
| | Receiver (S) | 380 (55) | 370 (37) | 488 (117) | 373 (76) |
| ELWI [ml kg$^{-1}$] | Donor | 12 (3) | 14 (3)* | 14 (2) | 14 (2) |
| | Donor (S) | 10 (1) | 10 (1) | 10 (1) | 12 (2) |
| | Receiver | 11 (2) | 10 (2) | 13 (1) | 15 (4)* |
| | Receiver (S) | 15 (8) | 15 (8) | 17 (10) | 18 (9) |
| CVP [mmHg] | Donor | 36 (66) | 6 (1) | 6 (2) | 19 (28) |
| | Donor (S) | 6 (2) | 6 (2) | 6 (2) | 6 (2) |
| | Receiver | 7 (2) | 13 (4) | 12 (8) | 8 (1) |
| | Receiver (S) | 8 (2) | 15 (1) | 10 (1) | 10 (1) |
| S$_p$O$_2$ [%] | Donor | 99 (1) | 93 (7) | 97 (2) | 97 (3) |
| | Donor (S) | 98 (1) | 98 (1) | 99 (1) | 99 (1) |
| | Receiver | 99 (1) | 98 (2) | 99 (1) | 98 (2) |
| | Receiver (S) | 98 (3) | 98 (1) | 99 (1) | 99 (1) |

Notes:

MAP, mean arterial pressure; HR, heart rate; PAP, mean arterial pulmonary pressure; CO, cardiac output; GEDI, global endiastolic volumen index; ELWI, entdiastolic lung water index; CVP, central venous pressure; SpO$_2$, oxygen saturation; (S), sham; BLH, baseline healthy; h, hours.

\* $p < 0.05$ vs. baseline value.

\# $p < 0.05$ in intergroup comparison.

**Table 2 Spirometric parameters.**

| Parameter | Group | BLH Mean (SD) | 0 h Mean (SD) | 4 h Mean (SD) | 8 h Mean (SD) |
|---|---|---|---|---|---|
| $_{et}CO_2$ [mmHg] | Donor | 40 (3) | 43 (3) | 40 (3) | 38 (2) |
| | Donor (S) | 39 (3) | 39 (3) | 39 (2) | 39 (1) |
| | Receiver | 40 (3) | 39 (2) | 41 (2) | 38 (2) |
| | Receiver (S) | 42 (1) | 44 (1) | 38 (1) | 36 (1) |
| $F_iO_2$ [%] | Donor | 41 (1) | 41 (3) | 41 (4) | 42 (3) |
| | Donor (S) | 40 (0) | 40 (0) | 40 (0) | 40 (0) |
| | Receiver | 40 (0) | 40 (1) | 40 (0) | 40 (0) |
| | Receiver (S) | 40 (0) | 40 (0) | 40 (0) | 41 (1) |
| $PaO_2/F_iO_2$ | Donor | 436 (35) | 180 (19)[*,#] | 256 (70) | 232 (54) |
| | Donor (S) | 451 (30) | 455 (35) | 453 (24) | 358 (46) |
| | Receiver | 488 (58) | 423 (90) | 356 (54) | 302 (67) |
| | Receiver (S) | 415 (54) | 410 (119) | 321 (89) | 281 (78) |
| $C_P$ [ml cmH$_2$O$^{-1}$] | Donor | 18 (2) | 14 (1) | 15 (1) | 15 (1) |
| | Donor (S) | 20 (1) | 20 (1) | 17 (1) | 16 (1) |
| | Receiver | 19 (2) | 15 (3) | 15 (2) | 14 (2) |
| | Receiver (S) | 19 (1) | 16 (3) | 15 (2) | 15 (2) |
| $R_{AW}$ [kPA l$^{-1}$ s$^{-1}$] | Donor | 14 (2) | 15 (2) | 14 (1) | 14 (1) |
| | Donor (S) | 15 (2) | 15 (2) | 15 (3) | 15 (3) |
| | Receiver | 12 (1) | 13 (1) | 13 (1) | 13 (1) |
| | Receiver (S) | 14 (1) | 15 (3) | 14 (2) | 15 (2) |
| FRC [ml] | Donor | 726 (115) | 622 (93) | 662 (123) | 649 (83) |
| | Donor (S) | 716 (368) | 716 (368) | 636 (314) | 670 (346) |
| | Receiver | 753 (215) | 641 (188) | 588 (180) | 613 (186) |
| | Receiver (S) | 920 (156) | 807 (35) | 579 (211) | 571 (154) |
| MV [l min$^{-1}$] | Donor | 7 (1) | 8 (1) | 8 (1) | 8 (2) |
| | Donor (S) | 7 (1) | 7 (1) | 8 (1) | 8 (1) |
| | Receiver | 7 (1) | 7 (1) | 7 (1) | 7 (1) |
| | Receiver (S) | 7 (1) | 8 (0) | 8 (1) | 8 (1) |
| TV [ml kg$^{-1}$] | Donor | 30 (14) | 41 (9) | 40 (8) | 28 (12) |
| | Donor (S) | 37 (7) | 37 (7) | 40 (10) | 41 (12) |
| | Receiver | 34 (9) | 18 (8) | 22 (8) | 28 (6) |
| | Receiver (S) | 30 (7) | 15 (1) | 24 (2) | 24 (3) |

(Continued)

| Table 2 (continued) | | | | | |
|---|---|---|---|---|---|
| Parameter | Group | BLH Mean (SD) | 0 h Mean (SD) | 4 h Mean (SD) | 8 h Mean (SD) |
| $P_{peak}$ [cm $H_2O$] | Donor | 18 (1) | 24 (3) | 22 (1) | 23 (2) |
| | Donor (S) | 17 (2) | 17 (2) | 19 (2) | 20 (2) |
| | Receiver | 17 (1) | 20 (4) | 20 (2) | 22 (2) |
| | Receiver (S) | 18 (1) | 20 (2) | 20 (1) | 21 (2) |
| $P_{mean}$ [cm $H_2O$] | Donor | 9 (0) | 12 (2) | 11 (1) | 11 (2) |
| | Donor (S) | 9 (1) | 9 (1) | 9 (0) | 10 (0) |
| | Receiver | 9 (1) | 10 (1) | 10 (1) | 10 (1) |
| | Receiver (S) | 9 (0) | 10 (1) | 10 (0) | 10 (0) |
| Peep [cm $H_2O$] | Donor | 4 (0) | 5 (2) | 5 (1) | 5 (2) |
| | Donor (S) | 4 (0) | 4 (0) | 4 (0) | 4 (0) |
| | Receiver | 4 (0) | 4 (0) | 4 (0) | 4 (0) |
| | Receiver (S) | 4 (0) | 4 (0) | 4 (0) | 4 (0) |

Notes:

$SpO_2$, oxygen saturation; $ex.CO_2$, expiratory carbon dioxide; $FiO_2$, fraction of inspired oxygen; $PaO_2/FiO_2$, oxygen index; Cp, pulmonary compliance; RAW, airway resistance; FRC, fraction of inspired oxygen; MV, minute volume; TV, tidal volume; Ppeak, peak inspiratory pressure; Pmean, mean airway pressure; PEEP, positive end-expiratory pressure; BLH, baseline healthy; h, hours; (S), sham.
* $p < 0.05$ vs. baseline value.
# $p < 0.05$ in intergroup comparison.

intergroup differences. Leukocyte count was significantly increased in the donor group after ARDS induction (4 h/8 h vs. BLH: $18.5 \pm 8.9/18.1 \pm 5.9$ vs. $10.6 \pm 3.4$; $p < 0.001$). After the blood transfusion, significantly higher haemoglobin and haematocrit counts were measured in the receiver group (a.e. haemoglobin 4 h vs. BLH: $11.1 \pm 0.9$ vs. $9.2 \pm 0.5$; $p < 0.05$). Further, a significant higher intergroup difference between the receiver and donor group was detected for haemoglobin (4 h/8 h: $11.1 \pm 0.9/10.5 \pm 0.8$ vs. $10.2 \pm 0.7/9.5 \pm 0.7$; $p < 0.001$). Table 3 highlights all the haematological parameters. After ARDS induction, increased serum levels of TNF-alpha were measured for the donor group without being significant (Fig. 3). Lactate levels between the donor and receiver groups were elevated (0 h: $3.0 \pm 1.6$ vs. $1.3 \pm 0.2$; $p < 0.001$). Furthermore, significant differences in lactate levels over time were observed in both groups ($p < 0.001$ donor group 0 h vs. BLH/4 h/8 h; $p < 0.05$ receiver group 8 h vs. 0 h/BLH; Table 4). Table 4 summarizes the blood gas analysis. The diffuse alveolar damage (DAD) score showed significantly higher levels in the ARDS donor group compared with all other groups ($p < 0.001$ donor group $29.35 \pm 6.38$ vs. receiver $19.50 \pm 5.79$/donor-sham $18.66 \pm 4.75$/receiver sham $18.91 \pm 5.12$). The lung wet-to-dry ratio showed no intergroup differences. Hippocampal mRNA expression for inflammatory response showed minimal increased levels for IL-6, iNOS and sLCN-2 in the receiver group compared to the donor group without being significant (IL-6: $1.77e-05 \pm 6.69e-06$ vs. $2.89e-06 \pm 1.09e-06$; iNOS: $2.28e-04 \pm 8.61e-05$ vs. $1.91e-04 \pm 7.20e-05$; sLCN-2: $2.27e-02 \pm 2.41e-02$ vs. $1.49e-02 \pm 1.01e-02$; $p > 0.05$; Fig. 4). No increased receiver hippocampal mRNA expression for TNF-alpha was measured

**Table 3 Haematological parameters.**

| Parameter | Group | BLH Mean (SD) | 4 h Mean (SD) | 8 h Mean (SD) |
|---|---|---|---|---|
| Leukocytes [$\mu l^{-1}$] | Donor | 10.6 (3.4) | 18.5 (8.9)* | 18.1 (5.9)* |
| | Donor (S) | 13.8 (5.9) | 14.3 (5.8) | 13.8 (4.2) |
| | Receiver | 17.5 (4.6) | 15.2 (5.5) | 14.2 (3.9) |
| | Receiver (S) | 17.7 (4.3) | 15.8 (1.3) | 13.7 (0.6) |
| Hemoglobine [$mg\ dl^{-1}$] | Donor | 9.5 (0.6) | 10.2 (0.7)* | 9.5 (0.7) |
| | Donor (S) | 9.7 (0.4) | 9.3 (0.4) | 9.0 (0.6) |
| | Receiver | 9.2 (0.5) | 11.1 (0.9)* | 10.5 (0.8) |
| | Receiver (S) | 9.1 (0.6) | 10.7 (0.7) | 10.2 (0.7) |
| Hematocrit [%] | Donor | 29.6 (2.1) | 31.6 (2.4) | 29.4 (2.1) |
| | Donor (S) | 30.7 (1.1) | 29.1 (1.3) | 28.1 (2.3) |
| | Receiver | 28.8 (1.7) | 33.6 (2.6)* | 32.5 (2.6)* |
| | Receiver (S) | 28.5 (1.6) | 32.7 (0.7) | 31.1 (1.9) |
| Thrombocytes [$\mu l^{-1}$] | Donor | 319 (58) | 230 (76) | 224 (82) |
| | Donor (S) | 388 (32) | 333 (39) | 318 (28) |
| | Receiver | 388 (55) | 268 (57) | 261 (52) |
| | Receiver (S) | 432 (51) | 337 (32) | 316 (17) |

Notes:
BLH, baseline healthy; h, hours; (S), sham.
\* $p < 0.05$ *vs.* baseline value.
\# $p < 0.05$ in intergroup comparison.

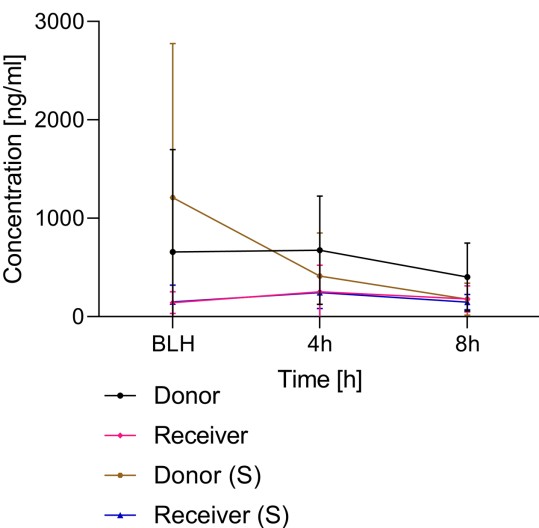

**Figure 3 TNF-alpha serum concentration.** BLH, baseline healthy; h, hours; (S), sham.

**Table 4 Blood gas analysis.**

| Parameter | Group | BLH Mean (SD) | 0 h Mean (SD) | 4 h Mean (SD) | 8 h Mean (SD) |
|---|---|---|---|---|---|
| Potasium [mmol l$^{-1}$] | Donor | 4 (0.2) | 4.4 (0.2) | 4.9 (0.6) | 5 (0.7) |
| | Donor (S) | 3.9 (0.2) | 3.9 (0.2) | 5.1 (0.5) | 4.3 (0.2) |
| | Receiver | 3.9 (0.2) | 3.9 (0.2) | 4.4 (0.2) | 3.8 (0.2) |
| | Receiver (S) | 4.1 (0.5) | 4.1 (0.4) | 4.6 (0.1) | 4.3 (0.1) |
| Lactat [mmol l$^{-1}$] | Donor | 0.9 (0.3) | 3 (1.6) [*,#,+] | 1.6 (1) | 1.1 (0.6) |
| | Donor (S) | 1 (0.4) | 1 (0.4) | 0.7 (0.2) | 0.6 (0.2) |
| | Receiver | 1.3 (0.3) | 1.3 (0.2) | 0.6 (0.1) | 0.4 (0.1)[*] |
| | Receiver (S) | 1.1 (0.3) | 2.6 (1) | 0.8 (0.1) | 0.5 (0) |
| $_{art}CO_2$ [mmHg] | Donor | 41 (3) | 50 (7) | 43 (6) | 42 (4) |
| | Donor (S) | 41 (2) | 41 (2) | 37 (1) | 37 (2) |
| | Receiver | 42 (3) | 39 (6) | 40 (3) | 39 (2) |
| | Receiver (S) | 43 (3) | 47 (4) | 37 (2) | 37 (1) |
| $PaO_2$ [mmHg] | Donor | 174 (14) | 75 (8) | 105 (26) | 95 (19) |
| | Donor (S) | 177 (9) | 177 (9) | 179 (10) | 141 (17) |
| | Receiver | 195 (23) | 169 (36) | 142 (22) | 121 (27) |
| | Receiver (S) | 171 (27) | 150 (46) | 129 (36) | 110 (28) |
| pH | Donor | 7.47 (0) | 7.35 (0) | 7.46 (0) | 7.47 (0) |
| | Donor (S) | 7.48 (0) | 7.48 (0) | 7.54 (0) | 7.53 (0) |
| | Receiver | 7.47 (0) | 7.43 (0) | 7.52 (0) | 7.52 (0) |
| | Receiver (S) | 7.48 (0) | 7.4 (0) | 7.57 (0) | 7.55 (0) |
| BE [mmol l$^{-1}$] | Donor | 4.5 (6.1) | 1.9 (3.2) | 6.1 (3.1) | 6.5 (3.0) |
| | Donor (S) | 6.3 (2.0) | 6.3 (2.1) | 8.6 (1.9) | 7.5 (1.5) |
| | Receiver | 6.2 (1.7) | 4.2 (2.8) | 9.1 (1.9) | 7.9 (1.6) |
| | Receiver (S) | 7.7 (0.6) | 4.5 (2.1) | 10.6 (0.5) | 9.8 (0.7) |
| $S_vO_2$ [%] | Donor | 57 (5) | 50 (20) | 56 (14) | 59 (4) |
| | Donor (S) | 58 (10) | 58 (10) | 54 (10) | 53 (8) |
| | Receiver | 57 (7) | 65 (7) | 59 (8) | 62 (4) |
| | Receiver (S) | 60 (15) | 65 (14) | 63 (11) | 63 (9) |

Notes:
$SvO_2$, central venous oxygen saturation; BE, base excess; art.$CO_2$, arterial carbon dioxide; $PaO_2$, arterial oxygen; (S), sham; BLH, baseline healthy; h, hours.
[*] $p < 0.05$ vs. baseline value.
[+] $p < 0.05$ vs. 4/8 h.
[#] $p < 0.05$ in intergroup comparison.

compared to the donor group. Higher mRNA expression of sLCN-2 in the cortex was measured without being significant (receiver vs. donor: 6.00e-03 ± 2.27e-03 vs. 3.97e-02 ± 1.50e-02; $p > 0.05$; Fig. 5). The other inflammatory markers in the cortex (IL-6, TNF-alpha,

## Hippocampus

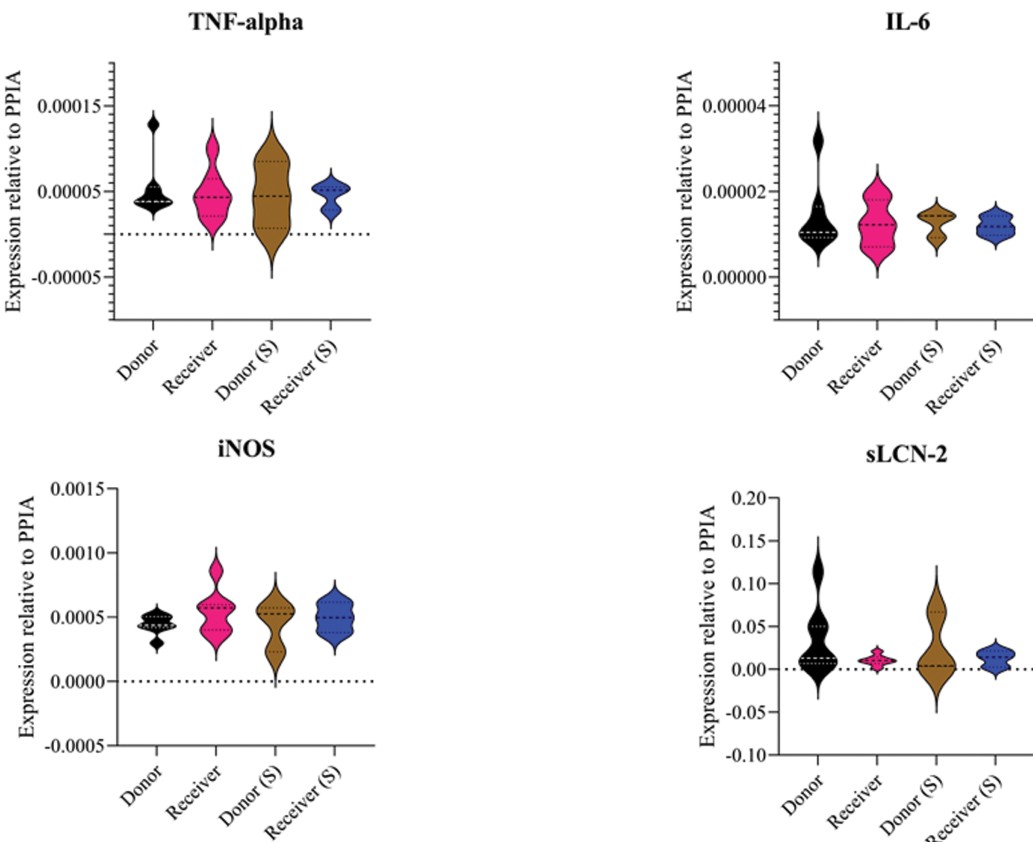

**Figure 4 Tissue sample analyses hippocampus.** Inflammatory marker expressions of interleukin-6 (IL-6), tumour necrosis factor alpha (TNF-alpha), inducible nitric oxide synthase (iNOS) and soluble lipocalin-2 (sLCN-2) in hippocampal tissue relative to PPIA expression. (S), sham.

iNOS) showed no increased mRNA expression in the receiver group compared to the donor group (Fig. 5). There were no significant changes in the associated sham groups.

## DISCUSSION

The present study investigated the roll of cytokine distribution after ARDS induction and the influence of the cerebral inflammatory response in a presumed interorgan cross-talk (*Pelosi & Rocco, 2011*; *Mrozek, Constantin & Geeraerts, 2015*). We found that blood sampling from ARDS donors in healthy pigs did not affect lung function or induce an inflammatory response in the brain or systemic circulation.

ARDS induction with oleic acid is a common model for the induction of pulmonary inflammatory response with consecutive pulmonary dysfunction. This model is highly standardised in our group. *Kamuf et al. (2018)* described the protocol and main advantages and disadvantages of ARDS induction with oleic acid. In the present study, ARDS was induced in the donor group in a sustained manner (Fig. 2). A minimal increase in TNF-alpha was measured as an early cytokine response marker of ARDS (*Herzum & Renz,*

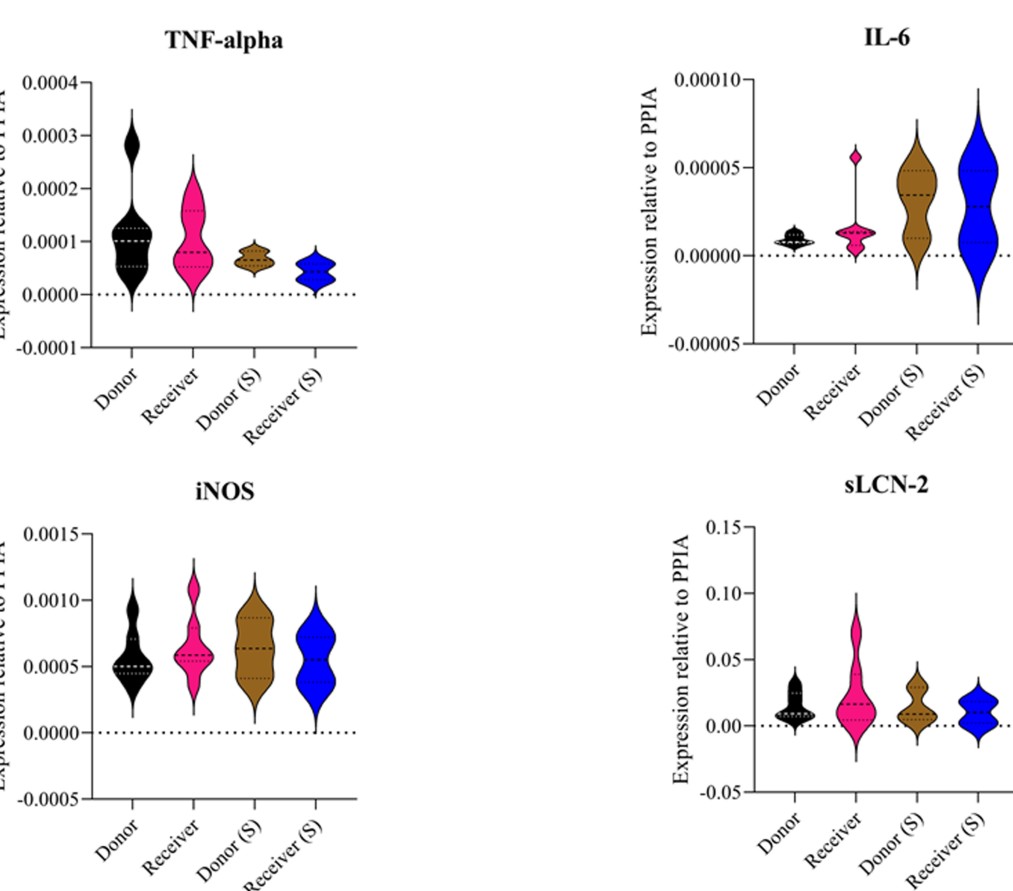

**Cortex**

**Figure 5 Tissue sample analyses cortex.** Inflammatory marker expressions of interleukin-6 (IL-6), tumour necrosis factor alpha (TNF-alpha), inducible nitric oxide synthase (iNOS) and soluble lipocalin-2 (sLCN-2) in cortical tissue relative to PPIA expression. (S), sham.

*2008*; *Butt, Kurdowska & Allen, 2016*). As a result of the inflammatory response, the alveolocapillary unit lost its integrity. First, the extra lung water index in the donor group increased significantly over time. Second, the postmortem analysis of the lung tissue confirmed lung damage in the ARDS donor group. These changes are known to occur early in the pathophysiological ARDS cascade and underscore the practicality of the ARDS model with oleic acid (*Butt, Kurdowska & Allen, 2016*; *Deja et al., 2008*). The 'transfusion of ARDS' to the receiver group was not as severe as might be expected and was not comparable to the extent seen in the donor group. At the postmortem analysis, no tissue changes were noted in the presumed ARDS blood receiver group.

Blood transfusions can be associated with serious consequences and, in the worst case, can lead to patient death (*Marik & Corwin, 2008*; *Goodnough, 2003*). The main causes behind this are the transfusion of incompatible blood, allergenic reaction, transfusion-related acute lung injury (TRALI) or transfusion-associated circulatory overload (TACO). Compared with human blood, porcine blood offers several advantages

(*Ziebart et al., 2019*; *Smith et al., 2006*). *Ziebart et al. (2019)* demonstrated that random allogenic blood transfusion in pigs was not associated with adverse effects of incompatibility, upregulation of systemic and local (brain/lung) inflammatory markers and did not affect lung function. Comparable to previous studies, there were no other differences in additionally measured parameters (*Ziebart et al., 2019*; *Bodenstein et al., 2012*). Allogenic blood transfusion in pigs is a safe approach to investigate the blood-bound transmission in remote organ injury, as claimed in our study.

The communication between the lung and brain is a complex structure. Three communication pathways have been described in the literature: a humoural, a cellular and a neural pathway (*López-Aguilar et al., 2013*). However, the descriptions of these pathways are not unambiguous. The activation of vagus nerve afferents in the neural pathway acts rapidly; other communication pathways are more time-consuming in this regard. Anti- and pro-inflammatory effects in the brain have been described (*Pavlov & Tracey, 2012*; *Tracey, 2002*). This is also known as 'inflammatory reflex'. A major stimulus is the activation of pulmonary mechanoreceptors during mechanical ventilation (*López-Aguilar et al., 2013*; *Quilez et al., 2011*). In the present study, this influence can be reduced by observing lung protective ventilation settings across the experiment. The cellular pathway is a medium-acting pathway. In particular, TNF-alpha increases monocyte chemotactic protein-1 (MCP-1) expression in the brain *via* the activation of TNF receptor 1 at the surface of microglia. The crossing of immune cells across the BBB is facilitated and multiplies the immune response in the brain (*D'Mello, Le & Swain, 2009*). The BBB plays a key role in the slow-acting humoural signalling pathway. The lung and brain share the same inflammatory markers as TNF-alpha. Once released by monocytes and macrophages in the primary affected organ (*i.e.*, lung), they can reach the brain and *vice versa*. As mentioned above, borderline elevated TNF-alpha levels were observed in the ARDS donor group in our study. Unfortunately, we did not see any significant cerebral inflammatory changes. This could be due the short duration of the experiment. At first, the BBB, as a protective firewall to maintain homeostasis for optimal brain function, is impermeable for many molecules, such as inflammatory cytokines and immune cells (*Quan & Banks, 2007*; *Quan, 2006*). The complex structure of this neurovascular unit enables unique functions: low exposure to systemic toxins and reduced traffic of inflammatory cells minimise local neuronal inflammatory damage (*Varatharaj & Galea, 2017*).

However, systemic inflammation causes changes in the BBB. A lipopolysaccharide (LPS) model of inflammation demonstrates that tight junction modification, glycocalyx degradation and astrocyte modulations cause BBB loss of function (*Minami et al., 1998*). High blood TNF-alpha levels have also been shown to reduce tight junction expression and contribute to BBB anatomic changes (*Tsao et al., 2001*; *Clausen et al., 2014*). This CNS effect occurs 12 h after Streptococcus pneumonia is injected into the bloodstream. Interestingly, this contrasts with the results of another LPS model. Intraperitoneally injected LPS causes a significant upregulation of IL-1ß and TNF-alpha in the hippocampus 3 h after injection (*Skelly et al., 2013*). Slow transmission through the circumventricular organs and choroid plexus has been described and is one way to bypass the BBB (*Dantzer,*

2001; *Konsman, Parnet & Dantzer, 2002*). The initiated upregulation of cortical and hippocampal inflammatory markers early in the process in this study could be interpreted as the nonsignificant minimally elevated levels of iNOS, IL-6 and sLCN-2. This upregulation of inflammatory markers contributes to the neurocognitive deficits observed in critically ill patients. After ARDS, patients have been found to have short-term confusion syndromes and long-term cognitive and memory impairments (*Hopkins & Jackson, 2006*; *Quilez et al., 2011*; *Hopkins & Brett, 2005*). A crucial key role is played by hypoxic and inflammatory changes in the area of the hippocampus (*Neves, Cooke & Bliss, 2008*).

Lipocalin-2 (LCN-2) is a protein that plays a role in various diseases and inflammatory cascades. LCN-2 is involved in the acute stress response, with multiple innate immune functions in a retinal degeneration model (*Parmar et al., 2018*). The protein initiates pro-survival pathways in the retina. Furthermore, LCN-2 is secreted by astrocytes under inflammatory conditions and acts as a chemokine inducer and promoter of the classical pro-inflammatory cascade (*Lee, Jha & Suk, 2015*). Upregulated expression of LCN-2 is also observed in mice with obesity and is responsible for neurological degeneration (*Jin et al., 2020*). Similar results have been reported in a rodent model. Increasing LCN-2 production is associated with decreased cognitive performance (*Pinyopornpanish, Chattipakorn & Chattipakorn, 2019*). In LPS-induced acute lung injury in mice, one of the most overexpressed genes in a serial gene analysis is LCN-2 (*Sun et al., 2005*). Comparable results have been reported for sepsis-related ARDS in humans. LCN-2 is a key mediator of the initial neutrophil response to the infection (*Kangelaris et al., 2015*). Therefore, our chosen study design should be able to mimic similar pathological stages and induce overexpression of LCN-2. One possible explanation why this is not the case could be that the role of LCN-2 during RBC transfusion has not yet been studied. LCN-2 plays an important role in iron homeostasis and as an antioxidant agent (*Xiao, Yeoh & Vijay-Kumar, 2017*; *Yamada et al., 2016*). Further studies that investigate the proposed pro- and anti-inflammatory effects of LCN-2 during transfusion are needed.

The present study has several limitations. First, the changes in cytokine activity and blood concentration in cold-stored red blood cells over 24 h remain unclear. There are no studies addressing this issue. It could be possible that the plasma concentration of donor cytokines shows fluctuations and inhibits the systemic distribution of ARDS in the recipient group. Due to local regulations, we couldn't do two attempts at 1 day, so the blood had to be stored. Second, as reported above for sepsis-related neurological changes after bacterial injection, the observation time in the present study may have been too short. CNS changes occur 12 h after the injection of Streptococcus pneumonia injections. We probably observed the first hours of the inflammatory cascade with the initiation of inflammatory gene expression. At the end of the experiment, we observed pulmonary deteriorations in the recipient group. This could be interpreted as a sufficient initiation of the proposed inflammatory cascade with signs of an acute lung dysfunction because of the 'ARDS transplantation' from the donor group. CNS changes were not seen at this time because of the shortness of the observation period.

## CONCLUSIONS

In conclusion, the chosen model of random allogenic red blood cell transfusion in pigs is feasible and reproducible without serious adverse events, as reported by *Ziebart et al. (2019)*. ARDS induction with oleic acid is a well-known method and was performed in the present study with success. Unfortunately, no ARDS-like alterations were seen in the recipient group, and CNS inflammation could not be detected. The role of the systemic distribution of inflammatory cytokines from one affected organ to another could not be confirmed beyond a doubt in the current study. Nevertheless, the circulation of cytokines in the bloodstream plays a major role in the process of organ cross-talk. In subsequent investigations, the observation period needs to be expanded. Furthermore, the role of cytokine transmission over the BBB and the other reported routes as a key mediator of CNS inflammation must be more clearly addressed.

## ACKNOWLEDGEMENTS

The study is part of the PhD thesis of Moritz Schaefer. The authors thank Dagmar Dirvonskis for the excellent technical assistance.

### Funding

The study was funded by a fellowship from the Mainz Research School of Translational Biomedicine (TransMed) affiliated with Johannes Gutenberg-University, Mainz, Germany. The funders had no role in study design, data collection and analysis, decision to publish, or preparation of the manuscript.

### Grant Disclosures

The following grant information was disclosed by the authors:
Mainz Research School of Translational Biomedicine (TransMed) affiliated with Johannes Gutenberg-University.

### Competing Interests

Erik Hartmann is an Academic Editor for PeerJ.

### Author Contributions

- René Rissel analyzed the data, prepared figures and/or tables, authored or reviewed drafts of the paper, and approved the final draft.
- Moritz Schaefer performed the experiments, prepared figures and/or tables, and approved the final draft.
- Jens Kamuf performed the experiments, authored or reviewed drafts of the paper, and approved the final draft.
- Robert Ruemmler performed the experiments, authored or reviewed drafts of the paper, and approved the final draft.

- Julian Riedel analyzed the data, authored or reviewed drafts of the paper, and approved the final draft.
- Katja Mohnke analyzed the data, authored or reviewed drafts of the paper, and approved the final draft.
- Miriam Renz analyzed the data, authored or reviewed drafts of the paper, and approved the final draft.
- Erik K. Hartmann conceived and designed the experiments, performed the experiments, prepared figures and/or tables, authored or reviewed drafts of the paper, and approved the final draft.
- Alexander Ziebart conceived and designed the experiments, performed the experiments, prepared figures and/or tables, authored or reviewed drafts of the paper, and approved the final draft.

## Animal Ethics

The following information was supplied relating to ethical approvals (*i.e.*, approving body and any reference numbers):

This prospective randomised animal study was conducted after approval by the State and Institutional Animal Care Committee (reference number: 23 177-07/G 14-1-084).

## Data Availability

The statistical analysis of the histological samples from the cortex and the hippocampus and all labor measurements are available in the Supplemental Files.

## Supplemental Information

Supplemental information for this article can be found online at http://dx.doi.org/10.7717/peerj.13024#supplemental-information.

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
