# Peer review of "Lung–brain ‘cross-talk’: systemic propagation of cytokines in the ARDS via the bloodstream using a blood transfusion model does not influence cerebral inflammatory response in pigs"

_PeerJ, doi:10.7717/peerj.13024_

## Round 0.1 · original submission · Major Revisions

Please address the concerns of both reviewers and revise the manuscript accordingly.

Reviewer 1 ·

Basic reporting

In this manuscript the authors study inter-organ cross talk between the lung and brain upon blood transfusion from pigs that suffered from acute respiratory distress syndrome. The authors are doing a great job addressing the limitations of the study and provide good future directions for this interesting research topic.
The manuscript would benefit from shorter sentences and more precise language for example the sentence line 58-60:
“What is an initial commonly induced inflammatory response because of brain or lung injury in one of these organs will frequently lead to secondary injuries in remote organs.”
Could easily been shortened and easier to read like this:
“Injuries to the brain or lung result in inflammatory responses, which frequently leads to secondary injuries in remote organs.”
The result section should be improved by putting the results from each figure into better context. In contrast the discussion section reads disjointed and would benefit from shortening and focusing on the immediate relevant aspects of the study. The authors should attach the raw data in a spreadsheet format so it is easily machine readable, in case other groups want to analyze data.
The data presented in figure 3 is hard to evaluate with the large overlapping error bars and points. A grouped bar graph or swarm plot would be more informative. To better evaluate the data in figure 4 and 5 each data point should be visible. Therefore swarm or violin plot is the appropriate way to present the data.
Minor comments:

On line 36 the authors use the acronym ARDS without once spelling out Acute respiratory distress syndrome. This can make it harder for the reader to quickly grasp the abstract.

Line 67, 69 the observations of “double-hit” and long-term effects described in the previous sentence need a reference

Line 157 what were the studies the group size was compared to a reference is missing

Line 201 - reference for the presumed inter-organ cross-talk is missing

Line 215 and 220 full stop punctuation is missing

Line 221 the abbreviations are not written out: Transfusion-related acute lung injury (TRALI)
Transfusion-associated circulatory overload (TACO)

Experimental design

The experimental design includes all necessary controls and the sample size appears reasonable. The experimental process is well described and references to the originally established protocol for inducing ARDS are given. The authors should better explain what motivated the storage of donor blood for 12 h, when it is unclear how it affects the cytokine activity (line 300). The statistical analysis is performed well, the data presentation could be improved (see basic reporting section).

Validity of the findings

The findings are robust and show that in the chosen experimental condition no ARDS alteration or CNS inflammation were detected in the recipient population. The conclusions are supported by the experimental findings. Furthermore the ARDS induction by oleic acid injection as well as the blood transfusion protocol was successfully reproduced in this study.

Reviewer 2 ·

Basic reporting

Major comment:
1. The author shows the result in a relatively pithy manner while discussing a lot in the discussion part. I prefer the author reorganize the results and discussion sections. As to each result, the author may describe in separate paragraph and make some comment and explanation at the same time. This will benefit the author a lot for understanding the whole manuscript.

Minor comment:
1) The resolution of the figure 1 is limited, please provide a clear one;
2) There is no figure legend for each figure and each table, please make a clear notification for each one;
3) The Donor (S) / Receiver (S) means Donor-sham/ Receiver-sham in this paper, but there is not any notification when is first time appeared;
4) The plot lines are all black in figure 2 figure 3, the error bars are superimposed. It is confusing for the readers to distinguish each other, please color them corresponding to different ones;
5) It’s better to make a full-name notification for those abbreviation, such as ARDS, PaO2, FiO2 et.al when they are mentioned at the first time, please check the whole manuscripts to figure it out.

Experimental design

The major concerns are:
1. In the result section, the author evaluates the expression mRNA level of all the inflammatory response markers in hippocampal and cortex, they are minimal difference between groups without being significant. The conclusion of this paper “No ARDS-like alterations were seen in the recipient group, and CNS inflammation could not be detected. The role of the systemic distribution of inflammatory cytokines from one affected organ to another could not be confirmed beyond a doubt in the current study.” is mainly based on these results. The mRNA is an indirect way for the expression level of those markers. The author should provide this value at a protein level, so ELISA assay is a direct and easy way to indicate this instead of mRNA level. At the same time, these cytokines at protein level of the lung may also be included as an internal control.

Validity of the findings

No comments

Additional comments

The manuscript by Rene et.al investigates the role of the systemic distribution of cytokines via the bloodstream in a porcine ARDS model for organ cross-talk and possible inflammatory changes in the brain. It was concluded that blood sampling from ARDS donors in healthy pigs did not affect lung function or induce an inflammatory response in the brain or systemic circulation.
ARDS is a life-threatening lung injury that allows fluid to leak into the lungs. The study of the cross-talk between lung and remote organs has profound significance. For example, the COVID-19 shares some characteristic with ARDS and study of brain-lung crosstalk in COVID-19 can provide experimental and clinical evidence and strategy for the treatment of infected patients.
I support the publication of this manuscript after the few major and minor comments mentioned above are revised.

---

## Round 0.2 · Minor Revisions

Please address remaining issues indicated by the reviewer and amend manuscript accordingly.

Reviewer 2 ·

Basic reporting

No problem

Experimental design

No problem

Validity of the findings

No problem

Additional comments

In the response of the authors: "In our study, we used the mRNA level in accordance to pre-studies with blood transfusion in pigs. But this is historical: actually we also use ELISA kits for detecting cytokines at the protein level in our running studies. "
So if the author have such data. Please indicate it in the results. if this is not in accordance to the mRNA. it's also okay to just make some explanation.

---

## Round 0.3 · accepted · Accept

All remaining concerns and suggestions were addressed and revised manuscript is acceptable now.